# Maresin1 Suppresses High-Glucose-Induced Ferroptosis in Osteoblasts via NRF2 Activation in Type 2 Diabetic Osteoporosis

**DOI:** 10.3390/cells11162560

**Published:** 2022-08-17

**Authors:** Zhanwei Zhang, Chonghao Ji, Ya-Nan Wang, Shiyue Liu, Maoshan Wang, Xin Xu, Dongjiao Zhang

**Affiliations:** 1Department of Implantology, School and Hospital of Stomatology, Cheeloo College of Medicine, Shandong University & Shandong Key Laboratory of Oral Tissue Regeneration & Shandong Engineering Laboratory for Dental Materials and Oral Tissue Regeneration, No. 44-1 Wenhua Road West, Jinan 250012, China; 2Department of Periodontology, School and Hospital of Stomatology, Cheeloo College of Medicine, Shandong University & Shandong Key Laboratory of Oral Tissue Regeneration & Shandong Engineering Laboratory for Dental Materials and Oral Tissue Regeneration, No. 44-1 Wenhua Road West, Jinan 250012, China

**Keywords:** maresin1, diabetic osteoporosis, NRF2, ferroptosis, glutathione peroxidase 4

## Abstract

Maresin1 (MaR1) is an endogenous pro-resolving lipid mediator produced from polyunsaturated fatty acids and is believed to have antioxidant and anti-inflammatory properties. The objective of this study was to estimate MaR1′s impact on type 2 diabetic osteoporosis (T2DOP) and its pharmacological mode of action. An in vitro high-glucose model of the osteoblast cell line MC3T3-E1 was constructed and stimulated with MaR1. Type 2 diabetic rats were used to establish in vivo models of calvarial defects and were treated in situ with MaR1. The results revealed that, aside from preventing mortality and promoting the osteogenic capacity of MC3T3-E1 cells, MaR1 increased nuclear factor erythroid-2 related factor 2 (NRF2) signaling as well as the activity of glutathione peroxidase 4 (GPX4) and cystine-glutamate antiporter (SLC7A11) and caused the restraint of ferroptosis under hyperglycemic stimulation. However, the therapeutic impact of MaR1 was significantly diminished due to NRF2-siRNA interference and the ferroptosis activator Erastin. Meanwhile, these results were validated through in vivo experiments. These findings imply that MaR1 activated the NRF2 pathway in vivo and in vitro to alleviate high-glucose-induced ferroptosis greatly. More crucially, MaR1 might effectively reduce the risk of T2DOP.

## 1. Introduction

Osteoporosis, fractures, and osteoarthrosis are a few complications resulting from type 2 diabetes mellitus (T2DM), a prevalent metabolic condition [1,2,3,4,5,6]. The systemic bone condition known as type 2 diabetic osteoporosis (T2DOP), which results in decreased bone mass and abnormalities in bone remodeling, is well-recognized. T2DOP also poses a significant risk for fractures [7], inadequate titanium implant osseointegration [8], and subsequent bone resorption [9]. T2DOP was linked to persistent hyperglycemia, advanced glycated end products, oxidative stress [10], and calcium and phosphorus metabolic problems [11], according to numerous earlier investigations. Currently, there is no specific effective therapy for controlling the progression of T2DOP. Hence, finding new therapeutic reagents will be of high significance.

It has been thought that T2DOP is attributable to several factors, including oxidative stress and the release of inflammatory factors [12]. Maresin1 (MaR1), a derivative of one of the primary omega-3 fatty acids (docosahexaenoic acid, DHA), is an endogenous pro-resolving lipid mediator produced mainly by macrophages. It has been demonstrated to significantly impact oxidative stress and inflammation [13,14,15]. MaR1 has been found to have strong protective effects on the joints [16], respiratory system [17], and neurological system [18]. Furthermore, it has also been confirmed to protect against cardiac injury, alleviate ulcerative colitis, and preserve the liver and renal system from injury, involving the up-regulation of NRF2 signaling [19,20,21,22,23]. In addition, MaR1 accelerates hard and soft tissue regeneration [24,25], though, to date, more research is still needed to determine the exact pathways by which MaR1 exerts its pharmacological effects.

The stress-inducible transcription factor nuclear factor erythroid 2-related factor 2 (NRF2), which controls the production of genes that combat oxidative stress, is a critical regulator of the cellular antioxidant response [26,27]. NRF2 is confined to the cytoplasm under physiological circumstances via its interaction with Kelch-Like ECH Associated Protein 1 (KEAP1), and KEAP1 mediates NRF2 ubiquitination for destruction, keeping NRF2 at low levels [28]. Under oxidative stress, KEAP1 is degraded via autophagy and releases NRF2. To up-regulate the activation of antioxidant genes, such as glutathione peroxidase 4 (GPX4), heme oxygenase-1 (HO-1), and others, free NRF2 is translocated to the nucleus, where it joins forces with antioxidant response elements (AREs) [29,30]. Numerous pathological problems, including cancer, diabetes, autoimmune diseases, gastrointestinal illnesses, and cardiovascular illnesses, are caused by imbalances in redox homeostasis [27]. It is becoming increasingly apparent how vital NRF2 transcriptional targets are for regulating lipid peroxidative reduction, free iron accumulation inhibition, and glutathione synthesis and metabolism, all of which delay the onset of the ferroptosis cascade [31].

As its name implies, “ferroptosis” is a type of cell death that depends on iron. Although iron is necessary for human life, studies in mice have shown that excess iron can cause bone loss, which is closely associated with osteoporosis [32,33,34]. Apoptosis, necrosis, and autophagy are biochemically, morphologically, and genetically separate from ferroptosis [35]. An excessive accumulation of iron-dependent lipid radicals is triggered by dysfunctional iron accumulation, lipid metabolism, and intracellular antioxidant components, including glutathione (GSH), GPX4, and others. This leads to the occurrence of iron-dependent cell death [36]. Its morphology also reveals abnormalities in the mitochondria, including condensation or swelling, an increase in membrane density, reduced or absent crista, and rupturing of the outer membrane [35]. According to reports, ferroptosis has been linked to pathological and malignant processes which cause a wide range of disease states [30]. Numerous investigations have established that oxidative stress contributes to the onset of diabetic osteoporosis [12,37,38]. Ferroptosis may nevertheless occur before the progression of diabetic osteoporosis, and its functions in this condition need to be further investigated.

As previously indicated, high glucose levels cause ferroptosis in T2DOP by elevating reactive oxygen species (ROS), lipid peroxidation, and glutathione depletion; more significantly, melatonin therapy is essential for alleviating T2DOP via the Nrf2/HO-1 pathway [39]. Additionally, it is well established that, in T2DOP, mitochondrial ferritin (FtMt) lessens oxidative stress brought on by an excess of ferrous ions, preventing osteoblasts from undergoing ferroptosis [40]. Therefore, the above studies have suggested the existence of ferroptosis in diabetic osteoporosis, which is associated with oxidative stress.

To investigate MaR1′s role in bone regeneration in T2DOP and the mechanisms that occur in bone under MaR1 treatment, this study used a T2DM model both in vivo and in vitro [41,42,43,44]. According to our research, MaR1 therapy altered the NRF2 pathway specifically for cystine–glutamate antiporter (SLC7A11)/GPX4 signals to protect osteoblasts from ferroptosis in T2DM so as to ameliorate osteogenesis in T2DOP.

## 2. Materials and Methods

### 2.1. Osteoblast Culture and Treatment

The osteoblastic cell line MC3T3-E1 was provided by the Cell Resource Center, Institute of Basic Medicine, Chinese Academy of Medical Sciences. Fetal bovine serum (FBS; Lonsera, Salto, Uruguay)-supplemented α-minimum essential medium (α-MEM; Biological Industries, Acre, Israel) was used to cultivate cells at 37 °C in a humidified atmosphere with 5% CO_2_. Cells were cultured in the appropriate osteogenic induction medium, which contained 10 nM dexamethasone, 50 mg/L ascorbic acids, and 10 mM β-glycerophosphate (Solarbio, Beijing, China).

MC3T3-E1 cells were cultured in various groups for 72 h to test the effects of MaR1, including Normal medium (α-MEM with 5.5 mM glucose), control medium (with the addition of 20 mM mannitol and palmitate vehicle to the Normal medium of 5.5 mM glucose; Sigma, Ronkonkoma, NY, USA), T2DM medium (25 mM glucose and 200 mM sodium palmitate to mimic diabetic conditions; Alladin, Shanghai, China), and T2DM medium with MaR1 (1 or 10 nM; Cayman, Ann Arbor, MI, USA). To investigate the existence and potential mechanism of ferroptosis in osteoblasts, the ferroptosis inhibitor ferrostatin-1 (Fer-1, 5 μM; MedChemExpress, Monmouth Junction, NJ, USA) and the ferroptosis activator Erastin (Era, 1 μM; MedChemExpress, Junction, NJ, USA) were administered to the cell cultures.

### 2.2. NRF2-siRNA Knockdown of Osteoblasts

NRF2-siRNA (GenePharma, Shanghai, China) transfects MC3T3-E1 cells using a Micropoly-transfecter (Micropoly, Shanghai, China) to knock down NRF2 gene expression. MC3T3-E1 cell were transfected for 24 h before being given the T2DM + MaR1 medium. SiNC was an siRNA negative control. Western blotting and immunofluorescence staining were used to confirm the effectiveness of transfection.

### 2.3. Cell Proliferation Assessments

#### 2.3.1. 5′-ethynyl-2′-deoxyuridine (EdU) Assay

The proliferation of MC3T3-E1 was tested using an EdU detection kit (RiboBio, Guangzhou, China). Following cell seeding and various treatments, EdU detection was performed on day three in a 48-well plate with 6 × 10^3^ cells per well. The following experimental procedures were in line with the article [45]. Finally, photographs of osteoblasts were recorded using fluorescence microscopy (Leica, Wetzlar, Germany). Additionally, the percentage of EdU-positive cells (red fluorescence) in total cells (blue fluorescence), representing the proliferative capacity of MC3T3-E1, was calculated using ImageJ software (National Institutes of Health, Bethesda, MD, USA).

#### 2.3.2. Cell Counting Kit-8 (CCK-8) Assay

Some 96-well plates with 3 × 10^3^/well cell density were used for sowing MC3T3-E1 cells. According to the experimental plan, the cells were treated differently when their fusion rate reached 70–90%. After three days of incubation, cell proliferation was evaluated using the Cell Counting Kit-8 kit (CCK-8, Dojindo Laboratories, Kumamoto, Japan), following the manufacturer’s instructions.

### 2.4. Quantitative Real-Time Polymerase Chain Reaction (qRT-PCR)

Reverse transcription was completed using the SPARKscript II RT Plus Kit (Sparkjade, Jinan, China), after total RNA was extracted using Trizol reagent (Sparkjade, China). Using 2 × SYBR Green qPCR Mix (With ROX) (Sparkjade, China), real-time PCR was performed on a LightCycler 480 real-time PCR machine (Roche, Basel, Switzerland). The exact primers (Azenta, Suzhou, China) were utilized to amplify the genes for Collagen I (*Coli*), alkaline phosphatase (*Alp*), runt-related transcription factor 2 (*Runx2*), and glyceraldehyde-3-phosphate dehydrogenase (*Gapdh*). Cycle threshold (Ct) values were utilized to compare relative mRNA expression, while *Gapdh* was used as an internal control. The experimental data were computed utilizing the 2^−ΔΔCt^ approach [46]. Each experiment was repeated three times.

### 2.5. Protein Isolation and Western Blotting

Following various treatments, total proteins were first extracted, and Western blotting was conducted. The experimental approach was previously mentioned in this article [47]. The primary antibodies used were as follows: anti-GPX4 (ab125066, Abcam, Cambridge, UK), anti-SLC7A11 (ab175186, Abcam, UK), anti-RUNX2 (ab236639, Abcam, UK), anti-NRF2 (CST12721T, Cell Signaling Technology, Danvers, MA, USA), anti-COLI (14695-1-AP, Proteintech, Beijing, China), anti-ALP (ET1601-21, HuaBio, Hangzhou, China), and anti-GAPDH (HRP60004, Proteintech, China). The secondary antibody (SA00001-2, Proteintech, China) was used at a 1:10,000 dilution ratio. Chemiluminescence reagents (Sparkjade, China) were used for detection, and ImageJ software (National Institutes of Health, Bethesda, MD, USA) was applied to calculate each band’s optical density. The relative content was determined and graphically displayed using the ratio between the protein of interest and GAPDH in the same sample.

### 2.6. Osteogenic Differentiation Assessments

#### 2.6.1. Alkaline Phosphatase (ALP) Staining

The osteogenic induction media was used to cultivate MC3T3-E1 cells (2 × 10^4^ cells/well) in 12-well plates for seven days while subjecting them to various treatments. A NBT/BCIP staining kit (Beyotime, China) was used to perform ALP staining, and the results were examined under a light microscope (Leica, Germany) to gauge the osteogenic differentiation capacity.

#### 2.6.2. Alizarin Red S (ARS) Staining

At a density of 2 × 10^4^ cells per well in 12-well plates, MC3T3-E1 cells were grown in an osteogenic induction medium for 14 days under various conditions. They were stained with 0.2% alizarin red solution (Solarbio, China) for 20 min after being fixed with 4% paraformaldehyde. After rinsing, the cells were examined under a light microscope (Leica, Germany).

### 2.7. Cells Immunofluorescence Staining

MC3T3-E1 cells were sown in 48-well plates at a density of 6 × 10^3^ cells/well for three days under various circumstances to determine the expression and distribution of NRF2. Cells were treated with primary anti-NRF2 antibodies (CST12721T, Cell Signaling Technology, Danvers, MA, USA) overnight at 4 °C after being fixed with 4% paraformaldehyde and blocked with 1% Albumin Bovine V (BSA, Solarbio, China) for an hour. The nuclei were stained with 4′,6-diamidino-2-phenylindole (DAPI, Solarbio, China) for 10 min after the cells were treated with an Alexa Fluor^®^ 594-conjugated goat anti-rabbit IgG secondary antibody (SA00013-4, Proteintech, China) for 2 h in the dark. Fluorescence microscopy (Leica, Germany) was used to take the pictures. ImageJ software was used to evaluate the mean fluorescence intensity of NRF2.

### 2.8. Measurement of Malondialdehyde (MDA) Levels

Each group of cellular proteins was extracted in accordance with the instructions provided with the Cell Malondialdehyde (MDA) Assay Kit (Colorimetric Method) (A003-4-1, Nanjing Jiancheng Bioengineering Institute, Nanjing, China). Each sample (0.1 mL) was combined with the MDA detection working solution (1 mL). After mixing, they were centrifuged at 4000 rpm for 10 min before being cooked for 40 min in boiling water. The MDA level was determined by measuring the absorbance at 530 nm with a microplate reader after adding 250 μL of supernatant to a 96-well plate.

### 2.9. Determination of Reduced Glutathione (GSH) Levels

According to the instructions for the GSH and GSSG Assay Kit (S0053, Beyotime, Shanghai, China), the cells were rinsed with phosphate-buffered saline (PBS, Biosharp, Beijing, China) and then centrifuged for collection. The precipitate and the protein removal reagent were combined, and then the mixture was centrifuged for 10 min at 3500 rpm. Utilizing a microplate reader and the supernatant, the reduced glutathione concentrations were ascertained. 

### 2.10. Intracellular Iron Ion Content Detection

The manufacturer’s recommendations were referred to while attempting to measure the intracellular iron ion concentration (Intracellular Iron Colorimetric Assay Kit, E1042, Applygen, Beijing, China). Cells were first lysed for two hours before being combined with a 4.5% potassium permanganate solution as needed. The iron ion detector was added after incubating the mixture for an hour at 60 °C. The samples were placed in a 96-well plate, and an optical microplate reader was utilized to measure the absorbance at 550 nm.

### 2.11. Determination of Lipid ROS Levels

In accordance with the experimental plan, osteoblasts were treated with various media while being seeded in 48-well plates (6 × 10^3^ cells/well). Following a fixation step with 4% paraformaldehyde, the cells were exposed to C11-Bodipy (Thermo Fisher Scientific, Waltham, MA, USA) for 15 min at room temperature. The images were taken using a microscope (Leica, Germany), and ImageJ software was utilized to determine the mean fluorescence intensity of C11-Bodipy Green, which gauges the level of lipid ROS in the cell.

### 2.12. Determination of Intracellular Reactive Oxygen Species (ROS) Levels

Cells were cultivated at a density of 5 × 10^4^ on 6-well plates using a Reactive Oxygen Species Assay Kit (Beyotime, China) and subjected to various treatments for 72 h. The culture medium was switched to a serum-free medium, and 10 mol/L of dichloro-fluorescein diacetate (DCFH-DA) was incubated for 30 min in the dark. Lastly, a flow cytometer was used to estimate the fluorescence intensity of the cells, which represented the levels of ROS in the cells, using an emission wavelength of 525 nm and an excitation wavelength of 488 nm.

### 2.13. Transmission Electron Microscopy (TEM)

Different intervention techniques were used to resuscitate MC3T3-E1 cells, transfer them to 6-well plates, and culture them for 72 h. The following procedures are consistent with those described in the article [45]. Ultrathin sections (50–70 nm) were sectioned and samples were stained sequentially with 2% uranyl acetate and lead citrate. The sections were placed under a transmission electron microscope (FEI F200C, Hillsboro, OR, USA) to observe the ultrastructure of cells and to obtain photographs.

### 2.14. Ethics Statement 

The Shandong University Hospital of Stomatology’s Animal Ethics Committee authorized the study’s methods (protocol code: 20220119). The entire procedure in our studies complied with ethical standards and was completed in accordance with the Shandong University Institutional Guidelines. The ARRIVE recommendations were followed when conducting animal experiments.

### 2.15. Experimental Animals and Models

The in vivo experiments included 20 male Sprague Dawley rats, which were randomly divided into four groups (n = 5 for each group): Normal group, T2DM group, T2DM + Vehicle group (gelatin sponge and saline), and T2DM + MaR1 group (gelatin sponge and MaR1). Figure A6 provides an illustration of the experimental procedure in vivo. Following one week of acclimatization feeding, diabetic rats were induced using a four-week high-fat, high-sugar diet (55.4% regular feed, 20% sucrose, 15% lard, 8% egg yolk powder, 1.5% cholesterol, and 0.1% sodium cholate; Pengyue Experimental Animal Center, Jinan, China). The diabetic model was completed by administering an intraperitoneal injection of streptozotocin (STZ; Solarbio, China) at a small dose of 30 mg/kg [44]. The Normal group also received a citrate buffer injection at the same time. Only rats with fasting blood glucose levels of more than 11.1 mmol/L and impaired insulin sensitivity were verified as diabetic after seven days and used in subsequent studies. Throughout the investigation, blood glucose levels and body weights were recorded weekly. Under isoflurane inhalation anesthesia, all rats underwent surgery with bilateral 5 mm diameter circular calvarial defects. MaR1 (10 μmol/L × 50 μL) was loaded in situ in a gelatin sponge and placed in the calvarial defect of the T2DM + MaR1 group. As a comparison, for the T2DM + Vehicle group, a gelatin sponge soaked in saline solution was used. The dose of MaR1 administration was determined by reference to previous studies [24,48]. The rats were euthanized using slow CO_2_ release in a watertight cage in the fourth week following the operation. The calvarial bone was dissected and removed and placed in a sterile fresh 4% formalin solution at 4 °C.

### 2.16. Sequential Fluorescence Labeling

Tetracycline and calcein (30 mg/kg each) were injected intraperitoneally on days 14 and 21 following the calvarial defect surgery to mark the bicolor fluorescence sequence of rat neoskeletal deposition activity. The bicolor fluorescence sequence labeling was observed on the calvarial bone using a laser scanning confocal microscope (Zeiss LSM880, Jena, Germany).

### 2.17. Microscopic Computed Tomography (MicroCT)

Microscopic computed tomography was used to evaluate bone healing four weeks after calvarial defect surgery (Quantum FX, Caliper Life Sciences, Hopkinton, MA, USA). A 5 mm round bone defect area was the region of interest (ROI), and the range for three-dimensional vertical reconstruction was the thickness of the skull. The ratio of bone volume to total volume (BV/TV) and the thickness and number of trabecular bones (Tb.Th and Tb.N) were computed to estimate osteogenesis. TV denotes the overall volume of space in the ROI, while BV denotes the volume of bone in the ROI.

### 2.18. Immunohistochemistry (IHC)

According to the procedures described in [44], a universal SP kit (ZSGB-bio, Beijing, China) and a DAB kit (ZSGB-bio, Beijing, China) were used to perform immunohistochemical staining to detect GPX4 (67763-1-lg, Proteintech, China) and NRF2 (16396-1-ap, Proteintech, China). Each sample was placed under a microscope (Olympus, Tokyo, Japan), with three fields of view chosen randomly. Using ImageJ analysis, NRF2 and GPX4 expressions were quantified, and the average density value for each view area was computed.

### 2.19. Statistical Analysis

All data are presented as the means ± standard deviations (SDs) of at least three separate experiments. The Shapiro–Wilk test was applied to ascertain whether the data were normally distributed. Two groups of normally distributed data were compared using Prism 8 software (GraphPad Software, La Jolla, CA, USA), and a one-way ANOVA was performed to investigate any significant group-to-group differences. Nonparametric tests were used for data that were not normally distributed. Statistical significance was defined as a *p*-value of less than 0.05. 

## 3. Results

### 3.1. MaR1 Improved Osteogenesis Inhibited by High Glucose in MC3T3-E1

In this investigation, 25 mM glucose and 200 mM palmitate were used to create an in vitro mimic diabetic state. We explored the proliferative and osteogenic differentiation capacities of the MC3T3-E1 cells in the T2DM medium to investigate whether the T2DM microenvironment had a sizable effect on various osteoblastic properties. T2DM-treated osteoblasts displayed a weaker proliferative capacity than the Normal group, according to the EdU fluorescence and CCK-8 assays (Figure 1A–C). The ability of osteoblasts to proliferate was somewhat restored with the supplementation of MaR1 to the T2DM medium, particularly at a concentration of 10 nM (Figure A5C). As assessed by analyzing the gray-value expression, Western blotting results showed that the T2DM medium caused a decrease in the expression of osteogenesis-related proteins (including COLI, ALP, and RUNX2), which the T2DM + MaR1 medium could rescue (Figure 1D,E). Following 7 and 14 days of treatment with an osteogenic induction medium, the T2DM + MaR1 medium consistently resulted in some recovery of ALP activity and ability to form mineralized nodules (alizarin red staining), demonstrating the T2DM + MaR1 group’s osteogenic property in comparison to the T2DM group (Figure 1F,G). As there was no statistical distinction between the Normal and Hyperosmolar control groups, increased osmolarity was not the reason for the reduction in proliferation and osteogenic differentiation (20 mM mannitol and 5 mM glucose), as Figure A1 shows. When taken as a whole, excessive hyperglycemia greatly reduced the ability of MC3T3-E1 cells to proliferate and differentiate into osteogenic tissues, while MaR1 could improve the osteogenesis processes. 

### 3.2. MaR1 Improved the Osteogenic Function of Osteoblasts in High-Glucose Medium by Activating NRF2 Signaling

Immunofluorescence and Western blotting experiments were conducted to delve deeper into the part that MaR1 plays in diabetes-induced osteoblast dysfunction. After the addition of MaR1, it was found that the expression of NRF2 in the whole cells and nuclei was significantly higher in the osteoblasts treated with a T2DM medium compared to the T2DM group, proving that MaR1 improved the expression of NRF2 overall and encouraged its transfer into the nucleus (Figure 2C,D). Additionally, NRF2 was deleted by transfection with NRF2-siRNA (small interfering RNA of NRF2), and immunofluorescence and Western blotting verified the efficacy of the deletion (Figure 2). 

As anticipated, compared to the T2DM + MaR1 + siNC group (small interfering RNA negative control), the positive benefits of MaR1 subsided once siRNA was used to knock down NRF2. The findings showed that the suppression of NRF2 reversed the improvement effect of MaR1 in terms of reduced osteoblast viability and differentiation capacity in the T2DM medium (Figure A2). On transfection with NRF2-siRNA, osteoblast COLI, ALP, and RUNX2 protein expression levels were down-regulated (Figure A2C,D), and ALP activity decreased (Figure A2E). In conclusion, a reduction in NRF2 exacerbated osteoblast dysfunction and limited MaR1′s ability to promote osteogenesis in high-glucose mimic conditions.

### 3.3. MaR1 Ameliorated High-Glucose-Induced Osteoblast Ferroptosis via NRF2 Activation

We have confirmed that MaR1 can activate NRF2 signaling and ameliorate high glucose-induced osteoblast viability decline and dysfunction, but the deeper mechanism remains unclear. In this study, we found that the ferroptosis inhibitor Fer-1 could reverse the adverse effects of the T2DM condition on osteoblast activity, which prompted us to propose that ferroptosis is involved in increasing high-glucose-induced osteoblast death (Figure A4A and Figure A5A). In addition, in order to confirm the evidence of the effects of Fer-1 and MaR1 alone on cell proliferation, we used the CCK-8 method and found that Fer-1 and MaR1 (1 or 10 nM) by themselves did not have any pharmacological effects on the viability of MC3T3-E1 cells in the Normal medium (Figure A4B). The iron-dependent cell death mechanism known as ferroptosis is caused by lipid peroxidation. The transcription factor NRF2′s downstream products, SLC7A11 and GPX4, are ferroptosis-specific proteins. To determine whether NRF2 promotes the proliferative activity and differentiation capacity of osteoblasts under high glucose conditions by remitting ferroptosis, we added the T2DM + MaR1 + Erastin group, Erastin being a promoter of ferroptosis. The concentration of Erastin was screened according to the representation in Figure A5B.

Cell viability was determined using the CCK-8 technique, and the findings demonstrated that Erastin and NRF2-siRNA could counteract MaR1′s therapeutic effects (Figure 3A). Reduced glutathione (GSH) is a crucial defense system for ferroptosis, and increased GSH content means that ferroptosis is lessened. The detection of GSH content in cells indicated that both a high-glucose environment and Erastin led to damage to the glutathione redox system. Similarly, the knockdown of NRF2 also disturbed the GSH system (Figure 3B). In addition, total iron content is one of the most critical factors in triggering ferroptosis. An iron ion kit was utilized to measure the intracellular iron ion content. The study revealed that the T2DM group had considerably more intracellular iron ions than the Normal group. MaR1 was used to restore the iron ion level when osteoblasts were cultured in T2DM medium, but NRF2 inhibition lessened this effect (Figure 3C). Lipid ROS were visualized using the Bodipy-C11 fluorescent probe, intracellular ROS levels were detected using flow cytometry, and lipid peroxidation levels were tested with the MDA kit. These vital studies highlighted that T2DM treatment significantly increased lipid peroxidation levels, which MaR1 might reduce. Lipid peroxidation and oxidative stress levels were consistently exacerbated in the NRF2-siRNA group and the Erastin group (Figure 3D–G). Western blot analysis showed that SLC7A11/GPX4 is a downstream NRF2 pathway. This was primarily demonstrated by the high expression of SLC7A11 and GPX4 in the T2DM + MaR1 group and the low expression of SLC7A11 and GPX4 in the T2DM + MaR1 + siNRF2 group, suggesting that MaR1′s ability to improve osteoblast proliferation and osteogenic differentiation depends on the inhibition of ferroptosis (Figure 3H,I).

Transmission electron microscopy can be used to observe mitochondrial ultrastructural changes, and the severe impairment of mitochondrial morphology is indeed associated with ferroptosis [49]. The mitochondria of the MC3T3-E1 cells were typically smaller, the membranes were stained more intensely, the numbers of cristae were decreased, and the structures of the folded membranes were disrupted in the T2DM group, T2DM + MaR1 + Erastin group, and the T2DM + MaR1 + siNRF2 group (Figure 3J). Overall, the main conclusion that can be drawn is that ferroptosis is involved in the pathologically abnormal bone metabolism caused by T2DM, which also provides a new avenue for studying the pharmacological mechanism and clinical application value of MaR1. 

### 3.4. MaR1 Protected Osteoblasts from Ferroptosis-Mediated Impairment of Osteogenic Differentiation by Modulating NRF2 Signaling in T2DOP

Considering that MaR1 affects ferroptosis in osteoblasts, a further study on osteogenic differentiation was conducted. EdU fluorescence results showed that treatment with a ferroptosis agonist (Erastin) could impair the cell proliferation activity of MC3T3-E1 in a high-glucose medium containing MaR1 (Figure 4A,B). Compared to the Normal group, the mRNA levels of the osteoblastic differentiation markers *Coli*, *Alp*, and *Runx2* were elevated in the T2DM + MaR1 group and the T2DM + MaR1 + siNC group. At the same time, they were lower in the T2DM group, the T2DM + MaR1 + Erastin group, and the T2DM + MaR1 + siNRF2 group (Figure 4C). The expression of the osteogenesis-related proteins COLI, ALP, and RUNX2 in the Western blot evaluation matched the qRT-PCR trend (Figure A3A,B). ALP staining further supported the results (Figure A3C). The above results proved that MaR1 ameliorated the high-glucose-induced inhibition of osteogenic differentiation through the NRF2-ferroptosis pathway, as the graphical abstract in Figure A9 shows.

### 3.5. MaR1 Promotes In Situ Bone Regeneration in a Calvarial Defect Model in Diabetic Rats

#### 3.5.1. MaR1 Enhances In Situ Bone Regeneration

Body weight, blood glucose levels, and oral glucose tolerance tests were used to assess the diabetic rat model. The figure displays the average body weight of each group. At the experiment’s start, all rats weighed 110 ± 20g. From day 7 after receiving a dose of streptozotocin (STZ), a statistically significant difference in body weight was observed between the Normal group and the other three diabetic groups, and the difference increased gradually (Figure 5A). All the rats had normal blood glucose levels at the beginning, but the diabetic group’s levels sharply rose on the seventh day following the intraperitoneal injection of STZ. Blood sugar was kept between 3.5 and 5.5 mmol/L in the Normal group and between 13 and 26 mmol/L in the three T2DM groups from day 7 after STZ injection (Figure 5B). The oral glucose tolerance test is a diagnostic test for diabetes. The rats were given a glucose solution (3 g/L). Blood glucose levels were checked for the rats in each group every 30 min, with the 120th minute serving as the final measurement time point. The study’s findings revealed that the blood glucose levels of the diabetic rats were higher than 11.1 mmol/L (Figure 5C). The illustration of the experimental design in vivo is shown in Figure A6.

In addition, the established type 2 diabetes rat model did not exhibit obvious damage to the islet β cells of the rat pancreas, and pancreas shape was still regular compared with the Normal group (Figure A7A). Liver cells in the three T2DM groups exhibited a small amount of steatosis, though they lacked any noticeable pathological alterations, such as cell edema and nuclear atypia, indicating that the local supplement of MaR1 did not adversely affect the rat liver in a harmful way. In the kidneys, the T2DM rats showed widening of the mesangial region, expanded and swollen glomeruli, and degraded tubular vacuoles. The renal pathological changes between the T2DM + MaR1 group, the T2DM group, and the T2DM + Vehicle group were not obviously different.

The three-dimensional reconstruction using micro-computed tomography (micro-CT) demonstrated that MaR1 contributed to the formation of the trabecular bone in skeletal defects. The T2DM group and the T2DM + Vehicle group (gelatin sponge and saline) had drastically reduced BV/TV, Tb.Th, and Tb.N values compared to the Normal group, according to the computational study’s findings. The fact that there was no distinction between the T2DM group and the T2DM + Vehicle group implies that the gelatin sponge alone was ineffective for bone repair and reconstruction in calvarial defects. Additionally, the T2DM + MaR1 group (gelatin sponge and MaR1) had higher BV/TV and Tb.N levels than the T2DM + Vehicle group. Tb.Th, however, showed no change (Figure 5D–F).

As shown in Figure 5G, in the bone defect area, the calcium deposition area was marked by tetracycline fluorescent dye (yellow) and calcein fluorescent dye (green), respectively, on the 14th and 21st days after surgery. The yellow and green marker scale increased significantly in the Normal and T2DM + MaR1 groups, indicating a more active osteogenic process, while only faint yellow spots appeared in the T2DM group (Figure 5G). In addition, consistent conclusions were reached from a computational analysis of mean fluorescence density (Figure 5H,I).

According to the results of the HE staining, the Normal group’s defect area was almost filled with new bone, but the defect areas of the T2DM group and the T2DM + Vehicle group were nearly filled with fibrous connective tissue without new bone. Surprisingly, sporadic new bone islands could be observed in the T2DM + MaR1 group (Figure 5J). A statistical analysis of the new bone formation area suggested that the new bone area of the T2DM + MaR1 group was dramatically more extensive than that of the T2DM group and the T2DM + Vehicle group, indicating that MaR1 had a beneficial impact on the repair of bone defects in diabetic rats (Figure 5K). 

Osteoporosis is the result of an imbalance in bone remodeling. By keeping a balance between osteoblasts and osteoclasts, bone remodeling maintains bone metabolic equilibrium [50]. MaR1 may have an inhibitory effect on osteoclast activity in diabetic bone defects, which can be determined by detecting osteoclast-related factors in the bone defect area and surrounding old bone margins, such as tartrate-resistant acid phosphatase (TRAP) staining. Unfortunately, we only observed sporadic rare TRAP-positive cells in each group of samples, with no statistical differences between them (Figure A8). In a previous study, MaR1-treated aged mice formed less cartilaginous callus at tibial fractures than the control group, which seemed to suggest that MaR1 treatment increased osteoclast activation, but in the treated and untreated groups, TRAP staining in wound tissue was similar [25]. However, it would be interesting to expand research on the function of MaR1 in osteoclasts in the future.

More importantly, MaR1 did not cause toxic damage to rat livers and kidneys, as shown in the HE staining in Figure A7. A hopeful element of treatment for type 2 diabetic osteoporosis is that the in situ administration of MaR1 increased bone growth in diabetic rats.

#### 3.5.2. MaR1 Enhances the Expression of NRF2 and GPX4

IHC staining sought to identify the expression of the NRF2 and GPX4 proteins as indicators for assessing ferroptosis and NRF2 signaling activation, respectively, in bone marrow cavities and the edges of old bone near defects. The images displayed the evaluation of GPX4 and NRF2 expression in various treatment groups (Figure 6A,C). In order to obtain more precise results, we performed an analysis of the expression of NRF2 and GPX4 in different treatment groups, and the results are shown in Figure 6B,D. Rats in the T2DM group and the T2DM + Vehicle group had decreased levels of NRF2 and GPX4 compared to Normal rats, confirming that under diabetic conditions the NRF2 redox system was disrupted and cellular ferroptosis occurred. In diabetic rats, the in situ application of MaR1 dramatically enhanced the expression levels of NRF2 and GPX4. These discoveries have further conclusively proven that MaR1 alleviates the T2DM-related deregulation of the NRF2-ferroptosis pathway.

## 4. Discussion

Osteoporosis is considered one of the significant complications of diabetes, and the current treatment methods are not adequate for treating osteoporosis caused by diabetes [51]. The mechanism of diabetic osteoporosis must be further investigated and viable pharmacological therapies must be developed. In the current work, we discovered that MaR1 could encourage NRF2′s overall expression and nuclear entrance. We also noted that ferroptosis, a novel cell-death pattern, had a role in the mechanism of diabetic osteoporosis. We discovered that MaR1 activated NRF2 signaling to efficiently reduce high-glucose-induced ferroptosis in osteoblasts, hence reducing osteoporosis. We showed that in situ application of MaR1 to promote bone formation may be one of the effective therapies for type 2 diabetic osteoporosis.

The results obtained with the best animal model for diabetic osteoporosis are currently inconclusive. In this study, after one month of feeding on a high-fat, high-glucose diet, rats were injected intraperitoneally with STZ (30 mg/kg) to mimic the pathological process of diabetes. On the seventh day after STZ injection, the rats in the three T2DM groups lost weight (Figure 5A), fasting blood glucose exceeded 11.1 mmol/L (Figure 5B), and insulin sensitivity decreased (Figure 5C), demonstrating the success of the diabetes model establishment [52,53]. On this basis, a calvarial defect model was established in order to observe diabetic bone regeneration [54]. One of the most often utilized surgical sites for studying in situ bone healing is that of the calvarial defect model, which is frequently employed in scientific studies. It is a standardized bone defect that is highly reproducible and can be utilized for histological and radiological investigation [55]. In particular, MaR1 was loaded on a gelatin sponge and promoted in situ osteogenesis in the calvarial defect model. The findings demonstrated that diabetes induces osteoporosis due to the fact that, on day 28 of healing, the BV/TV percent in the T2DM group was markedly smaller than in the Normal group (Figure 5D). 

Additionally, 25 mM glucose and 200 mM sodium palmitate were utilized in this study to imitate the in vitro diabetic milieu for osteoblasts [41,42], and we discovered that this combination greatly decreased the activity and osteogenesis potential of the MC3T3-E1 cells (Figure 1). A cell type commonly utilized in bone tissue engineering is the MC3T3-E1 mouse pre-osteoblast cell line, which was developed from the calvaria of C57BL/6 mice. It exhibits attractive properties, such as a pro-osteoblast phenotype, high proliferation and differentiation capacity, Collagen I synthesis on day 3 of the culture period, and ALP enzymatic activity during the culture period. The MC3T3-E1 cell line is appropriate for researching the osteogenesis mechanism in vitro because it has straightforward culture conditions, a defined source, and stable properties [56]. In conclusion, both in vivo and in vitro models of T2DOP have been developed with success.

MaR1, according to early studies, is a family of naturally occurring anti-inflammatory, pro-resolving mediators found in human tissue fluids. Maresins have been proven in numerous studies to have critical preventive effects against a variety of diseases [15,57,58]. These lipid media have unique receptors with a smaller physiologically dynamic range than other protein molecules, often at nano- to dermal molar concentrations. Based on the results of former studies, the peak efficacy of MaR1 treatment for MC3T3-E1 cells may be 1 to 10 nM in vitro, and the maximal effective dosage in vivo needs to be 10 to 100 times higher than that in vitro [24]. The breakdown, release, and pharmacokinetics of MaR1, which was loaded through a gelatin sponge in animal models, should be further studied.

More recent works have demonstrated its antioxidant properties in diabetic patients [59,60]. However, there is no evidence that MaR1 reduces the incidence of osteoporosis in T2DM. Our study is the first to emphasize that MaR1 ameliorates bone density and treats T2DOP, especially the in situ application of MaR1 in calvarial defects in type 2 diabetic rats. 

Notably, the restraint in the expression of NRF2 was abrogated upon MaR1 treatment in T2DM, which could, in principle, enhance the expression of a variety of antioxidants. The crucial transcription factor NRF2 is necessary to maintain intracellular redox homeostasis and control cellular oxidative stress [61,62]. NRF2 can control oxidative stress in cells and promote the expression of antioxidant proteins [63]. NRF2 occurs in the cytoplasm and is connected to KEAP1 to keep NRF2 at low levels under typical physiological circumstances. NRF2 separates from KEAP1 in response to external stimuli, and after moving to the nucleus it attaches to the promoter region and activates downstream molecules, including NAD(P)H quinone oxidoreductase 1 (NQO1), HO-1, GPX4, SLC7A11, and others [64]. If excessive ROS causes peroxidation, the antioxidant system collapses, and the cellular suicide procedure is initiated, with ferroptosis drawing more attention. We thus looked further into whether MaR1 activated NRF2 to treat high-glucose-induced osteoblast dysfunction. We discovered that MaR1 stimulates the expression of NRF2 and the osteogenesis process in both in vitro (Figure 2 and Figure A2) and in vivo (Figure 6) investigations. The protective effect of MaR1 was significantly reduced when NRF2 was knocked down (Figure A2). Therefore, we hypothesized that MaR1 achieves its osteogenic role by activating NRF2.

Among the downstream target molecules of NRF2, we found that GPX4 and SLC7A11 are essential markers of ferroptosis as well. An increasing number of studies have recently revealed that ferroptosis is involved in the mechanisms underlying diabetic osteogenesis [65]. As a pattern of regulated cell death, ferroptosis is distinguished by intracellular iron overload and lipid ROS accumulation [66]. According to this study, the ferroptosis inhibitor ferrostatin-1 (Fer-1) could reverse the effect of the T2DM medium on osteoblast activity, which prompted us to propose that ferroptosis allows for increased high-glucose-induced osteoblast death (Figure A4A). We therefore investigated the possibility that ferroptosis occurred under high-glucose conditions, which would be rescued by MaR1. 

Mitochondria are considered to be a prime site of ferroptosis [67]. Transmission electron microscopy images confirmed abnormal changes in the mitochondrial morphology of osteoblasts under high-glucose circumstances, including the rising density of mitochondrial membranes and the loss of mitochondrial crest degradation. In contrast, MaR1 protected mitochondria from the aggression of high glucose (Figure 3J). In addition, MaR1 mitigated the increases in intracellular iron ion concentration, the accumulation of intracellular ROS and lipid oxides, and the damage to the SLC7A11/GSH/GPX4 axis—a critical ferroptosis defense system in cells. Under adverse stimuli, GSH redox system depletion occurs, resulting in the inactivation of GPX4 and increased intracellular lipid peroxidation and ferroptosis (Figure 3). Moreover, ferroptosis inhibitor (Fer-1) and MaR1 (1 or 10 nM) alone did not cause cell proliferation (Figure A4B). 

These in vitro results are in agreement with the in vivo tests. In type 2 diabetic rats, MaR1 was observed to enhance the level of NRF2 and GPX4 protein expression (Figure 6). More surprisingly, the combination of MaR1 and gelatin sponge assisted the bone tissue regeneration process at the defect site, which would eventually degrade in situ and be replaced by newly formed bone tissue (Figure 5). Overall, these findings imply that elevated glucose levels cause ferroptosis in osteoblasts and that MaR1 is essential for preventing this kind of ferroptosis.

Notably, a limitation of this study is that it did not provide direct evidence of ferroptosis. The antioxidant ferroptosis inhibitor Fer-1 does not specifically prevent ferroptosis and does not directly show that ferroptosis is occurring in cells, despite having a protective impact (Figure A4A). Similarly, decreased GPX4 activity, increased iron levels, and lipid peroxidation could not indicate that ferroptosis was directly involved. In fact, oxidized phospholipid (OxPL) species are the primary indicators of ferroptosis. Through lipidomic analysis, the measurement of ferroptotic OxPL species [68,69,70], and the imaging of OxPL using gas cluster ion beam–secondary ion mass spectrometry [71], only a few studies have revealed direct evidence of ferroptosis. Previous research using OxPL lipidomic analysis demonstrated that OxPL species accumulation is a hallmark of ferroptosis [70]. This study’s limitations are the absence of lipidomic data on OxPL levels as well as quantified ferroptosis-related factors in mitochondria, such as acyl-CoA synthetase long-chain family member 4 (ACSL4) [72] and 15-lipoxygenase (15-LOX) [73,74].

Apparently, ferroptosis is not the only mechanism of cell death connected to diabetic disease [75]. Through hyperglycemia, increased glycation end products, and exacerbated inflammation, diabetes disrupts bone metabolism and causes osteoblasts to undergo apoptosis, resulting in a decrease in osteoblasts and an increase in osteoclasts, impairing bone production [76]. In diabetes-related organ lesions, pyroptosis and necroptosis have also been observed, which may be potential mechanisms for the death of normally functional cells [76,77,78]. They may be involved in the pharmacological effects of MaR1, though this remains to be explored.

We have confirmed that MaR1 dramatically ameliorated the suppression of osteogenesis in T2DOP. We have investigated the potential parts that MaR1 could play in type 2 diabetes-induced ferroptosis in osteoblasts (Figure A9). Further studies, in light of the work provided here, are needed to complete our understanding of how MaR1 activates the expression of NRF2, of other pathways involving MaR1, and of the extent to which MaR1 alleviates T2DOP by inhibiting ferroptosis.

## 5. Conclusions

In summary, we have shown that T2DOP with high glucose levels could impair the biological characteristics of osteoblasts, which exhibited excessive oxidative stress and lipid peroxidation. More critically, this research enabled the conclusion that MaR1 could ameliorate high-glucose-induced ferroptosis and alleviate poor osteogenesis in T2DOP through NRF2 activation. This is an important finding in understanding the mechanisms related to MaR1 in promoting osteogenesis in T2DM, suggesting vital therapeutic targets for poor osteogenesis in type 2 diabetic patients.

## Figures and Tables

**Figure 1 cells-11-02560-f001:**
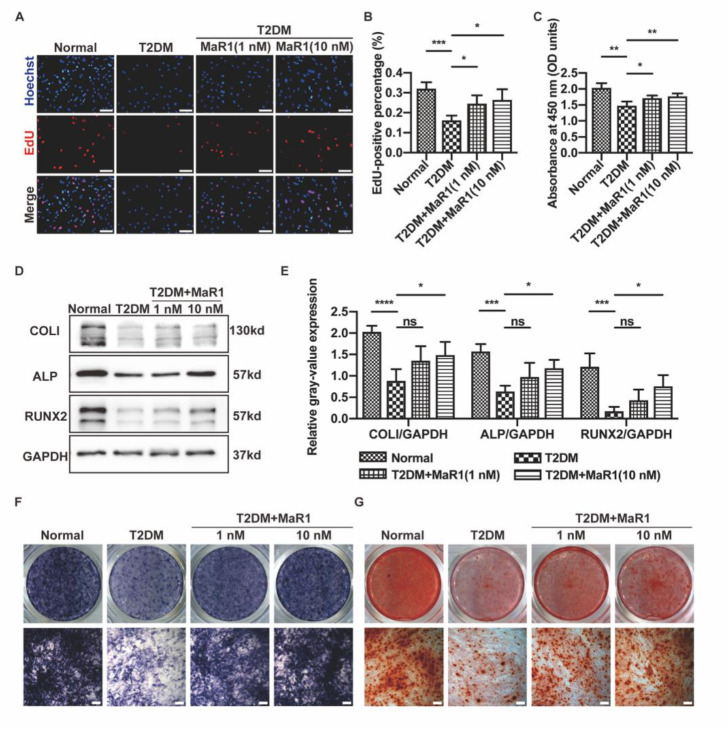
MaR1 improved the proliferation and osteogenic capability of osteoblasts in T2DM conditions. (**A**) To assess the osteoblasts performing DNA replication, EdU staining was tested. EdU-positive nuclei (red) and all nuclei (blue). Scale bar = 100 μm. (**B**) The quantitative study of cells using EdU staining. (**C**) At the end of the second day of incubation, the Cell Counting Kit-8 assay was conducted. (**D**) After seven days of osteogenic induction, Western blot analysis was used to evaluate the protein levels of COLI, ALP, and RUNX2. (**E**) The study of the Western blot analysis quantification of gray-value expression. (**F**) ALP staining following a week of osteogenic induction Scale bar = 200 μm. (**G**) ARS staining following a 14-day osteogenic induction. Scale bar = 200 μm. Data are presented as means ± SDs from three separate experiments (n = 3). ns *p* > 0.05; * *p* < 0.05; ** *p* < 0.01; *** *p* < 0.001; **** *p* < 0.0001.

**Figure 2 cells-11-02560-f002:**
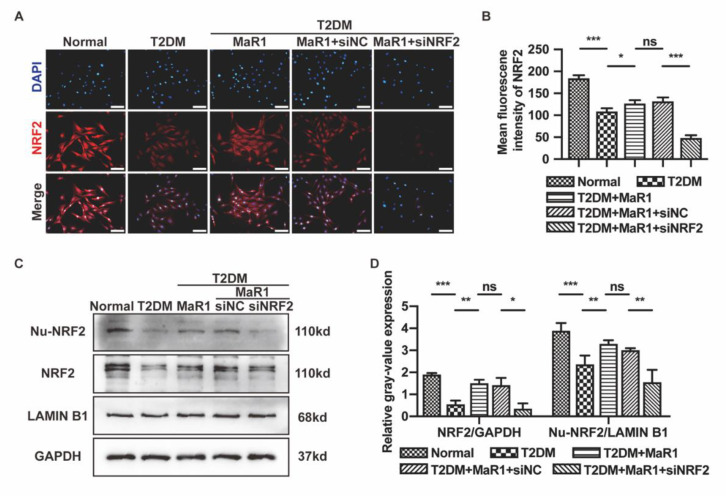
MaR1 activated NRF2 signaling under T2DM conditions. (**A**) After three days of various stimulations, the expression of NRF2 was examined using double immunofluorescent labeling. NRF2 proteins (red) and nuclei (blue). Scale bar = 100 μm. (**B**) The quantitative measurement of the average NRF2 fluorescence in osteoblasts. (**C**) Western blotting was used to assess the protein levels of NRF2 in nuclei (Nu-NRF2) and entire cells (NRF2). (**D**) The quantitative analysis of the Western blotting gray-value expression. Data are presented as means ± SDs from three separate experiments (n = 3). ns *p* > 0.05; * *p* < 0.05; ** *p* < 0.01; *** *p* < 0.001. SiNC: siRNA negative control; siNRF2: siRNA of NRF2.

**Figure 3 cells-11-02560-f003:**
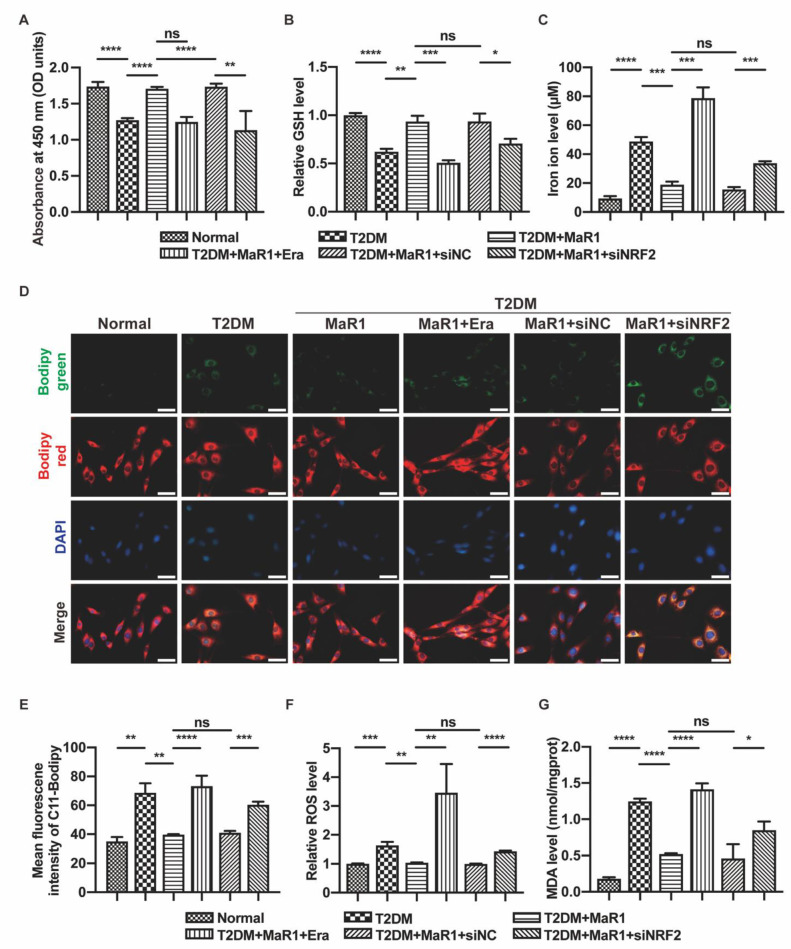
MaR1 ameliorated high-glucose-induced osteoblast ferroptosis via NRF2 activation. (**A**) The survival of the osteoblasts was examined using the Cell Counting Kit-8 assay. (**B**) Level of intracellular GSH. (**C**) Iron ion content. (**D**) The lipotropic dye, the Bodipy-C11 fluorescence probe, was used to identify lipid ROS. Green, oxidized lipids; red, normal lipids; blue, all nuclei. Scale bar = 50 μm. (**E**) The quantitative analysis of mean Bodipy-C11 fluorescence intensity. (**F**) Flow cytometry was used to detect the levels of intracellular ROS. (**G**) A malondialdehyde assay was used to measure lipid peroxidation. (**H**) Western blot analysis was used to determine the protein levels of GPX4 and SLC7A11. (**I**) Protein level quantification and analysis. (**J**) Images of MC3T3-E1 cells after various treatments were taken using transmission electron microscopy. Altered mitochondria are indicated by arrows. Scale bars = 500 nm. Data are presented as means ± SDs from three independent experiments (n = 3). ns *p* > 0.05; * *p* < 0.05; ** *p* < 0.01; *** *p* < 0.001; **** *p* < 0.0001. SiNC: siRNA negative control; siNRF2: siRNA of NRF2.

**Figure 4 cells-11-02560-f004:**
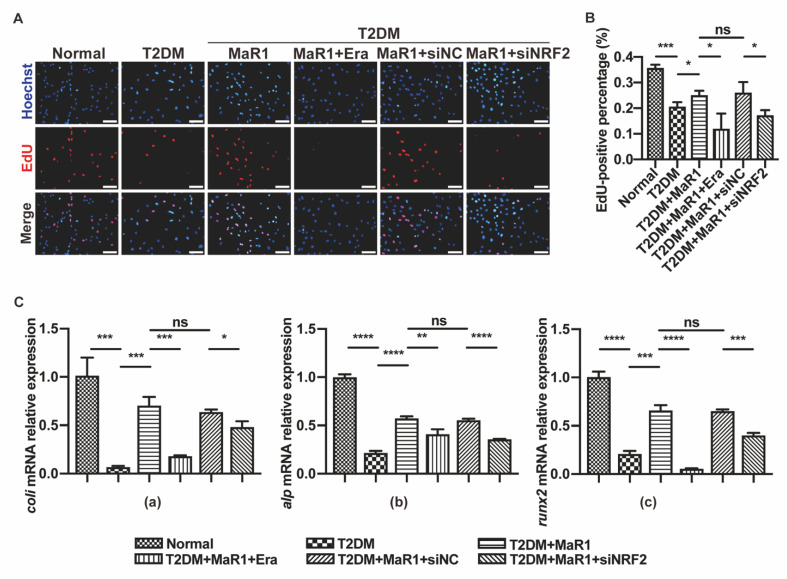
MaR1 protects from ferroptosis-mediated impairment of osteogenic differentiation of osteoblasts by modulating NRF2 signaling in T2DOP. (**A**) EdU staining was used to assess cell survival. Red, EdU-positive nuclei; blue, all nuclei. Scale bar = 100 μm. (**B**) EdU-positive cell quantification analysis. (**C**) The mRNA expressions of *Coli*, *Alp,* and *Runx2* were detected using quantitative real-time polymerase chain reaction (qRT-PCR). Data are presented as means ± SDs from three separate experiments (n = 3). ns *p* > 0.05; * *p* < 0.05; ** *p* < 0.01; *** *p* < 0.001; **** *p* < 0.0001. SiNC: siRNA negative control; siNRF2: siRNA of NRF2.

**Figure 5 cells-11-02560-f005:**
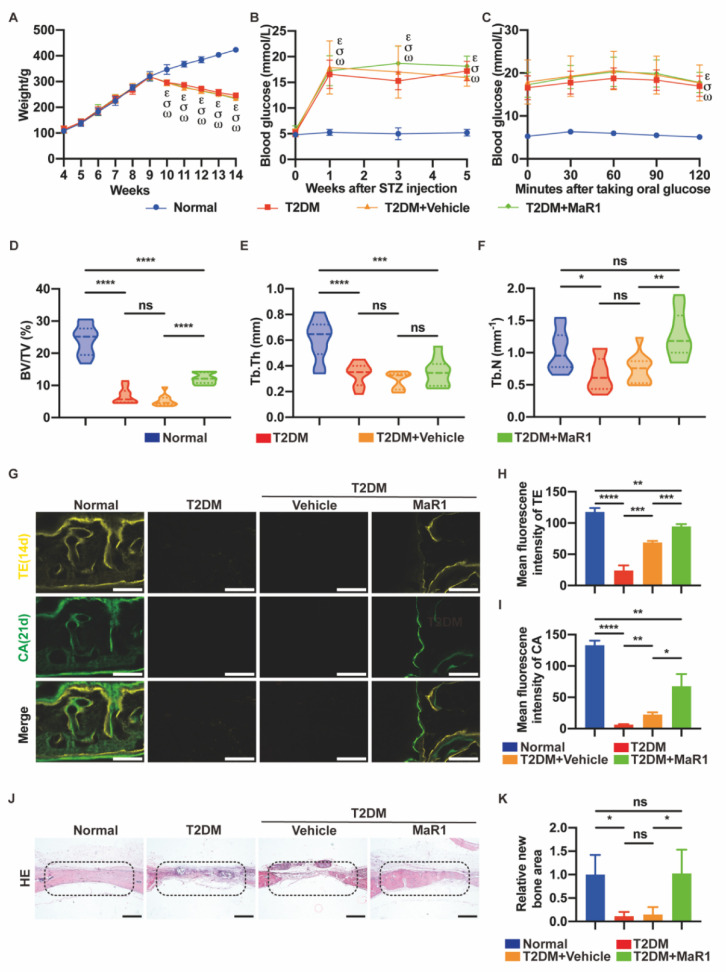
MaR1 promoted in situ bone regeneration in a calvarial defect model in diabetic rats. (**A**) Weight and (**B**) blood glucose in the four groups. (**C**) Fasting blood glucose levels after taking oral glucose in different experimental rats. ε: *p* < 0.05, for T2DM group vs. Normal group; σ: *p* < 0.05, for T2DM + Vehicle group vs. Normal group; ω: *p* < 0.05, for T2DM + MaR1 group vs. Normal group. (**D**) Three-dimensional reconstructed micro-CT of bilateral 5 mm rat calvarial defects was performed one month after surgery. Quantification analysis of BV/TV%, (**E**) Tb.Th mm, and (**F**) Tb.N mm^−1^ using micro-computed tomography. (**G**) Bicolor fluorescent labeling in the hard tissue slicing of the rat skulls. TE, tetracycline (yellow); CA, calcein (green). Scale bar = 100 μm. (**H**) A quantitative analysis of TE and (**I**) CA mean fluorescence intensity one month after surgery. (**J**) HE staining demonstrated the development of new bone in the defects. Black dotted frame: bone defect area. Scale bar = 500 μm. (**K**) Quantification study of the new bone development region in HE staining. Defects with a 5 mm diameter were indicated by a black dotted frame. Data are presented as the means ± SDs; n = 3 specimens/group; ns *p* > 0.05; * *p* < 0.05, ** *p* < 0.01, *** *p* < 0.001, **** *p* < 0.0001. Normal: normal diet and citrate injection; T2DM: four-week high-carbohydrate, high-fat diet, with STZ intraperitoneal injection (30 mg/kg); T2DM + Vehicle: gelatin sponge was placed in the defect, and the rest were the same as the T2DM group; T2DM + MaR1: gelatin sponge soaked in MaR1 was placed in the defect, and the rest were the same as the T2DM group. BV, bone volume; TV, total volume; Tb.Th, thickness of trabecular bone; Tb.N, number of trabecular bones.

**Figure 6 cells-11-02560-f006:**
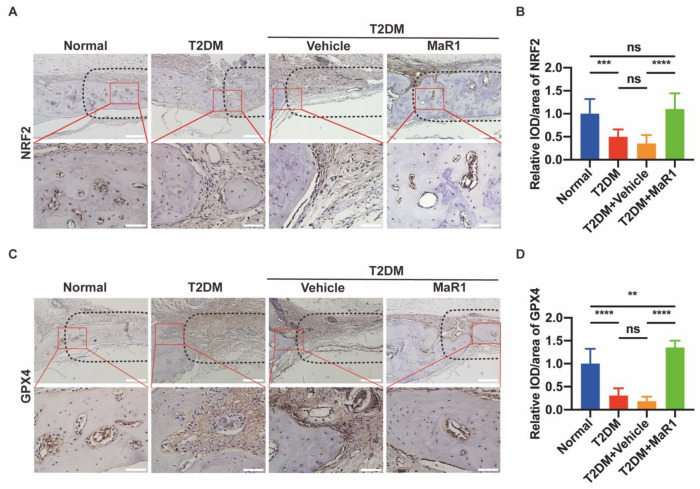
MaR1 enhances the expression of NRF2 and GPX4. (**A**) NRF2 protein immunohistochemistry staining. Scale bar = 200 μm and 50 μm. (**B**) A quantitative examination of NRF2′s mean integral optical density. (**C**) GPX4 protein immunohistochemistry pictures. Scale bar = 200 μm and 50 μm. (**D**) Mean integral optical density quantification analysis for GPX4. Data are presented as the means ± SDs for three separate specimens/groups (n = 3). ns *p* > 0.05; ** *p* < 0.01; *** *p* < 0.001; **** *p* < 0.0001. The black dashed box represents the defect area, and the red solid line box is the magnified visual field. IOD: the mean integrated optical density in the newly formed bone area and the edge of the old bone near the defect. Normal: Normal diet and citrate injection; T2DM: four-week high-carbohydrate, high-fat diet, with STZ intraperitoneal injection (30 mg/kg); T2DM + Vehicle: gelatin sponge was placed in the defect, and the rest were the same as the T2DM group; T2DM + MaR1: gelatin sponge soaked in MaR1 was placed in the defect, and the rest were the same as the T2DM group.

## Data Availability

Not applicable.

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
