# Peer review of "Maresin1 Suppresses High-Glucose-Induced Ferroptosis in Osteoblasts via NRF2 Activation in Type 2 Diabetic Osteoporosis"

_cells, 2022, doi:10.3390/cells11162560_

Round 1

Reviewer 1 Report

The manuscript submitted by Z. Zhang and co-workers and entitled “Maresin1 Suppresses High Glucose-induced Ferroptosis in Osteoblasts via NRF2 Activation in Type 2 Diabetic Osteoporosis” presents a fairly interesting study that refers to the investigation of estimation of the impact of Maresin1, a macrophage-derived anti-inflammatory and pro-resolving lipid mediator, on the type 2 diabetes-induced osteoporosis (T2DOP) which is a severe chronic complication caused by diabetes that affects the skeletal system. The authors created an in vitro mimic diabetic state to investigate the effect on various osteoblastic properties in the MC3T3-E1 cells using the T2DM model. The results imply that ferroptosis is involved in the mechanism of diabetes-induced osteoporosis, and Maresin1 is essential for preventing this kind of ferroptosis machinery by affecting the NRF2 signaling pathway. The authors also showed that Maresin1 enhances the level of GPX4 expression, which is the master regulator of ferroptosis. Overall it is an interesting study on an important topic that regards the involvement of ferroptosis in the T2DOP. I recommend the paper for publication. However, I have a few suggestions/concerns, presented below, which may improve the quality of this manuscript:

  • I would suggest to give less technical details in the abstract (Lines 18-19, 21) and improve the conclusion section by providing more clearly described conclusions.
  • The manuscript’s figures seem to be overloaded. Some of the panels or figures could be placed in the Supplementary Material. Also, some experiments are providing similar evidence, therefore, they also could be moved to the Supplementary Material, for example:

The expression of the osteogenesis-related proteins COLI, ALP, and RUNX2 in the western blot evaluation matched the qRT-PCR trend (Figure 5D and E). ALP staining further supported the results (Figure 5F).”   Lines 403-405.

·        Moreover, for some of the results the authors could provide more details or explanations, for example, MaR1+siNC – is used in Fig. 3 but it was not mentioned until Fig. 5 (Line 401), and the meaning was not explained at all in the text. Also, the description for Fig. 7 is very limited (here also the meaning of the “Vehicle” was not provided). Fig. 7 contain four panels and none of them was used in the main text, the same situation as Fig. A2.

  • The manuscript would benefit if the Authors would expand/change some sentences to make them more clear (in the Results section), for example:

o   it may eventually lead to bone loss or lost   Line 41-42

o   “Iron accumulation, lipid metabolism, and intracellular anti-oxidant components, such as glutathione (GSH), glutathione peroxidase 4 (GPX4), and others, are dysfunctional, causing excessive accumulation of iron-dependent lipid radicals, triggering the occurrence of iron-dependent cell death[34].”      Line 75-78

o   “.. GSH content are opposite to the degree of ferroptosis”    Line 352-353 (degree of ferroptosis – strange expression)

o   “.. improving ferroptosis ..”     Line 347

·        Missing references should be added:     T2DM model     - Line 96
                                                                To the statements at lines 51, 59 and 88,

·        Maresin1 is a polyunsaturated fatty acid  (Line 15).

·        “Ferroptosis”  (line 81), “Scale bar” (line 490) should be written with small letters;  

·        Line 453: Figure 6L doesn’t exist.

·        I am wondering, is there any particular reason why the authors didn’t mark SLC3A2 in Fig. A4?

  • Authors might be interested to take a look at the publication Ref1 where authors studied Maresin1 effect on Nrf2/HO-1/GPX4 pathway but in liver injury.
  • Did the authors consider the involvement of lipoxygenases in this process? Fig. A2A suggest some effect of Fer-1 in the T2DM conditions. It has been proved that Fer-1 is 15-lipoxygenase (in a complex with PE-binding protein 1) inhibitor (Ref2). 15LOXs are giving the biggest contribution to the generation of lipid hydroperoxides (+ Fenton reaction -> ferroptosis), etc.

Ref1: Maresin1 Protect Against Ferroptosis-Induced Liver Injury Through ROS Inhibition and Nrf2/HO-1/GPX4 Activation by W. Yang et al., Frontiers in Pharmacology, 2022.

Ref2: Resolving the paradox of ferroptotic cell death: Ferrostatin-1 binds to 15LOX/PEBP1 complex, suppresses generation of peroxidized ETE-PE, and protects against ferroptosis by T. Anthonymuthu et al., Redox Biol, 2021.

Maresin1 Protect Against
Ferroptosis-Induced Liver Injury
Through ROS Inhibition and Nrf2/
HO-1/GPX4 Activation
Wenchang Yang 1†, Yaxin Wang 2†, Chenggang Zhang 1
, Yongzhou Huang 1
, Jiaxian Yu 1
,
Liang Shi 1 , Peng Zhang 1
, Yuping Yin 1 , Ruidong Li 1
* and Kaixiong Tao 1

Reviewer 2 Report

The study is interesting and important but requires significant revisions to clarify details of experiments, add limitations and expand the discussion of the results that could improve the quality, readability, and significance of the manuscript:

1. What is the sample size (biological repeats) per each group in Fig. 5? Sample sizes should be indicated for all figures, and details of the experiments should be given for each figure in Methods and Figure legends.

2. Line 228: Experimental model of diabetes should be described in detail (a simple scheme of experimental protocols can be added). The sex of the rats should be shown. What is the rationale for using the animals in a large range of sugar levels (5-18 mmol/l) (line 241)? Any gravimetric data (e.g., body weight, etc) of diabetic rats vs. control?  

3. Figure 4: What means “Obsortance” in the Y-axis (Pabel A, the same in Fig. A2)? How were normalized GDH levels in Panel B (umol/L of what?). Representative WB images on panel H don’t match with quantitative data on panel I.

4. Erastin was used at a very low concentration (5 nmol). Have the authors performed dose-dependent experiments for erastin and Fer-1 in osteoblasts?

5. All known anti-ferroptotic compounds (Fer-1, Lip-1) are not specific inhibitors of ferroptosis; they are antioxidants, and the cardioprotective effects of these compounds do not provide direct evidence of the existence of ferroptosis in the cell (Effects of Fer-1 in Fig. A2). Likewise, reduction of GPX4 activity, increased iron levels, and lipid peroxidation do not indicate a direct involvement of ferroptosis. The main markers of ferroptosis are oxidized phospholipids (oxPL) species. Only a few studies analyzed and quantified ferroptotic PLox species by detailed lipidomics (PMID: 33824345; PMID: 27842066; PMID: 34102574) and visualized PLox by gas cluster ion beam secondary ion mass spectrometry (PMID: 33684237) thereby, providing direct evidence of ferroptosis.

6. Likewise, ultrastructural changes observed in MC3T3-373 E1 cells don’t “directly reflect the occurrence of ferroptosis” (Line 372-379).  Ferroptotic PEox species as well as ferroptosis-related factors such as ACSL4, GPX4, 15LOX (lipoxygenase) were not quantified in mitochondria. In previous studies, ferroptosis was confirmed in mitochondria isolated from tissue and cultured cells by lipidomic analyses of PLox (PMID: 34102574). The lack of lipidomic data on oxidized PL levels should be mentioned as a limitation of the study at the end of Discussion. Also, it should be mentioned that high glucose could stimulate not only ferroptosis (PMID: 34258295), but other programmed cell death mechanisms (apoptosis, necroptosis, pyroptosis, etc.) (PMID: 35813208; PMID: 35427562; PMID: 32092332).   

Reviewer 3 Report

cells-1827990

-Some redundancies are present in the introduction section sentences. Please, carefully check and amend them.

-The protocol number of the study acceptance issued by the Ethics Committee must be included in sub-item 3.14.

-The total number of mice used in the study should be provided. Replace the word sacrificed by euthanized. Moreover, the methods used for euthanasia (anesthesia and etc.) and sample collection should be described in detail.

-What did the authors use to define the dose? In the absence of data in the literature on the ideal dose for the model, the most appropriate would be to perform a dose-response experiment. Furthermore, another study that evaluated the analgesic effect of Mar-1 used a much lower dose (10 ng) when compared to the present study dose. Why was this single, extremely high dose selected? The authors should have in mind the translation to humans and reflect whether the present dose would be attainable and safe in humans.

Finally, the toxicity of larger organs such as liver and kidney must be measured, and the data presented to compose the safety of the dose used.

-Line 509: Unlike other bone defect models, the skull defect model is less affected and is the gold standard…….reference is needed.

-It would be very productive to perform additional experiments in diabetic rats, including the evaluation of osteoclastogenic factors including TNFalpha, and RANK/RANKL signaling. The role of osteoclasts in the model could not be neglected. These experiments will improve the quality of data and strengthen the conclusion of the study.

Round 2

Reviewer 2 Report

In response to reviewers' comments, the authors extensively revised the manuscript and thus, improved its presentation, readibility, and significance. 

Author Response

Dear Editor,

Thank you for your valuable and helpful comments concerning our manuscript (cells-1827990, Maresin1 Suppresses High-Glucose-induced Ferroptosis in Osteoblasts via NRF2 Activation in Type 2 Diabetic Osteoporosis). Special thanks to you for your work concerning our paper.

Wish you all the best!

Sincerely,

Dongjiao Zhang

Reviewer 3 Report

The Reviewer did not identify the presentation of the results of Figure A8 in the main text of the manuscript. The figure and its respective legend only appears at the end of the manuscript .

The manuscript is ready for acceptamce after checking this point. 
